# OpenReview forum: "VoiceAgentBench: Are Voice Assistants ready for agentic tasks?"
_ICLR.cc/2026/Conference — ICLR 2026 Conference Withdrawn Submission_

### Official Review · Reviewer_fPsi · 2025-10-31

**Soundness:** 2
**Presentation:** 2
**Contribution:** 2
**Rating:** 4
**Confidence:** 4

**Summary:**

This paper introduces VoiceAgentBench, a new benchmark for SpeechLLM agents. The benchmark covers various tool-calling scenarios and safety evaluations. It also covers several languages beyond English. The evaluation reveals significant limitations of current SpeechLLMs as agents, providing some insights into this direction.

**Strengths:**

- The topic is quite timely in the era of AI agents.
- Well-scoped problem & broad coverage.
- Layered evaluation design. The split into Tool Selection, Tool-Call Structure, Parameter-Filling, and Refusal Rate gives interpretable failure modes, instead of a single pass/fail.

**Weaknesses:**

- Synthetic speech only: Whether the synthesized speech is of good quality is unknown. There is also no consideration of real-world noisy environments.
- Lacking details of LLM-judge, such as decoding strategies and evaluation qualtiy (compared with human annotation).
- Metric design choices may bias outcomes. Tool Selection is an exact-match on function name; Tool-Call Structure hinges on Pydantic schema validation. These can over-penalize semantically correct outputs with minor format variance, while PF then depends on an LLM judge. A fuller error taxonomy or tolerance bands (e.g., argument similarity) beyond the judge would strengthen conclusions.
- The coverage of the evaluated models is quite restricted. The paper should include more models for a more comprehensive overview.
- No confidence intervals or significance tests for the results.

**Questions:**

- Why was GPT-4o-mini chosen as the sole judge for PF/RR? Did you try alternative judges? Any evidence of judge–model bias?
- In ASR→LLM pipelines, did you test stronger Indic ASR? Can you quantify how much of the Indic gap remains after substituting better ASR models? This will provide better insights than directly using ground-truth transcriptions.
- Can you provide details in the one-shot settings?

---

> ### Author Response · Authors · 2025-11-21
>
> We thank the reviewer for the thoughtful and positive assessment. We appreciate the acknowledgment of the timeliness of our topic, the clarity of our problem scope, and the breadth of our coverage. Our detailed responses to all comments are provided below.
>
> >**W1:** Whether the synthesized speech is of good quality is unknown. There is also no consideration of real-world noisy environments.
>
> We thank the reviewer for raising this point. To ensure the quality of the synthesized speech, we selected **ElevenLabs** TTS for English and **Krutrim** TTS for Hindi based on human-evaluated **Mean Opinion Scores (MOS)** computed on standard open-source datasets. For English, we conducted evaluations on LJSpeech; for Hindi, we used the IISc SYSPIN corpus.
>
> Each TTS system was rated across four standard perceptual dimensions: *naturalness, prosody, pronunciation, and clarity*. We compared ElevenLabs, Google TTS, Sarvam TTS, and Krutrim TTS using 50 samples per language, and selected the models that consistently achieved top MOS performance in our human study.
>
> **1. English TTS MOS Evaluation (50 Samples, LJSpeech)**
>
> | **Dimension**   | **Google TTS** | **ElevenLabs TTS** | **Sarvam TTS** | **Krutrim TTS** |
> |-----------------|----------------|---------------------|----------------|------------------|
> | **Naturalness** | 1.9            | **4.44**                | 3.66           | 4.16             |
> | **Prosody**     | 4.0            | **4.38**                | 4.3            | 4.24             |
> | **Pronunciation** | 4.64         | **4.72**                | 4.08           | 4.68             |
> | **Clarity**     | **4.7**            | 4.54                | 3.84           | 3.78             |
>
> **2. Hindi TTS MOS Evaluation (50 Samples, IISc SYSPIN)**
>
> | **Dimension**     | **Google TTS** | **ElevenLabs TTS** | **Sarvam TTS** | **Krutrim TTS** |
> |-------------------|----------------|---------------------|----------------|------------------|
> | **Naturalness**   | 2.24           | **3.84**                | 3.76           | 3.20             |
> | **Prosody**       | 2.34           | 3.56                | 3.06           | **3.60**             |
> | **Pronunciation** | 3.08           | 3.70                | 3.64           | **3.72**            |
> | **Clarity**       | 3.46           | **3.96**                | 3.86           | **3.96**             |
>
> For English (LJSpeech dataset), Eleven Labs achieved average MOS scores of approximately 4.44 (naturalness), 4.38 (prosody), 4.72 (pronunciation), and 4.54 (clarity), while for Hindi (IISc SYSPIN dataset), Krutrim TTS scored roughly 3.20, 3.6, 3.72, and 3.96 in the same dimensions. This evaluation guided our choice of TTS engines. We have added the details of this pilot study in Appendix N.
>
> While we did not explicitly simulate noisy real-world environments, our speech generation includes diversity-based speaker sampling covering a wide range of speaking styles to ensure meaningful variability in speech characteristics. This design allows the benchmark to capture realistic agentic performance under varied clean-audio conditions. In future versions, we plan to incorporate noise, reverberation, and other real-world acoustic variations to further stress-test the robustness of speech-agent systems.
>
> We have also prepared a lightweight demo page that provides category-wise speech examples from VoiceAgentBench: https://anonymous.4open.science/r/vab-2833/Examples.md.

---

> > ### Author Response · Authors · 2025-11-21
> >
> > >**W2 & Q1:** Choice of GPT-4o-mini as LLM-Judge, its details and reliability.
> >
> > ### Details of LLM-Judge:
> >
> > Our evaluation setup uses GPT-4o-mini as the LLM-judge for both parameter-filling and refusal-rate assessment. Since exact string matching is brittle for tool arguments, the judge is prompted to (i) reason step-by-step about whether the model’s predicted arguments semantically match the gold intent and (ii) output a binary verdict. For safety evaluation (refusal rate), we follow the procedure of AgentHarmBench [1], using GPT-4o-mini as a semantic refusal classifier.
> >
> > All judge prompts, reasoning formats, and input templates are fully documented in Appendix J.2 for easy reproducibility. Decoding uses the standard OpenAI API with default settings (temperature = 1.0); no custom decoding strategies are employed.
> >
> >  [1] AgentHarm: A Benchmark for Measuring Harmfulness of LLM Agents [ICLR 2025]
> >
> >
> > ### Human Agreement Study for reliability:
> >
> > To assess the reproducibility and human alignment of our GPT-4o-mini–based evaluation for parameter-filling accuracy, we conducted an additional human agreement study on a representative subset of our benchmark. We find significant agreement between GPT judge and human responses (with average **0.92** pairwise agreement while kappa average of **0.79**).
> >
> > **Experimental setup:** We sampled ~200 instances across categories and languages and collected ~400 model responses from KimiAudio and Whisper-Llama3.3-70B. Each response was labeled as correct or incorrect by two independent expert annotators (Annotator 1 and Annotator 2). We additionally compute a human-majority label, defined as the consensus between the two annotators for agreement analysis. We evaluate alignment using two complementary metrics where we have substantial agreement:
> >
> > **1. Pairwise Agreement:**
> >
> > This measures raw agreement rates between the GPT-4o-mini judge and human annotators.
> >
> > - GPT-4o-mini vs. Annotator 1: 0.9039
> > - GPT-4o-mini vs. Annotator 2: 0.9360
> > - GPT-4o-mini vs. Human Majority: 0.9236
> > - Annotator 1 vs. Annotator 2: 0.9680
> >
> > **2. Cohen’s Kappa:**
> >
> > We additionally report Cohen’s Kappa to assess reliability of GPT-4o-mini as a judge.
> >
> > - GPT-4o-mini vs. Annotator 1: 0.7488
> > - GPT-4o-mini vs. Annotator 2: 0.8310
> > - GPT-4o-mini vs. Human Majority: 0.7953
> > - Annotator 1 vs. Annotator 2: 0.9141
> >
> > Considering these scores and the cost–effectiveness of GPT-4o-mini, we consider it a justified choice for serving as the LLM judge in our framework. We have added the details and results of this experiment in the Appendix M and have updated Section 3.3 to mention this study.
> >
> > >**W3:** Metric design choices may bias outcomes. These can over-penalize semantically correct outputs with minor format variance, while PF then depends on an LLM judge.
> >
> > Our evaluation metrics were intentionally selected to isolate distinct failures in tool-using agents.
> >
> > First, **JSON structure and tool-name correctness** are evaluated with strict binary criteria because these components are prerequisite for any successful tool invocation: an otherwise well-reasoned response becomes unusable if the structure is invalid or if the wrong tool is selected. Similarly, incorrect tool selection can lead to degraded  user experiences. For these reasons, we adopt exact matching for these components, as minor deviations in these fields are not semantically interchangeable in real systems.
> >
> > In contrast, **parameter filling** often admits a range of acceptable variants that may differ lexically but are semantically equivalent in practice (e.g., *“New York City”* vs. *“NYC,”* or natural-language dates vs. canonical timestamp formats). To accommodate this inherent flexibility while maintaining rigorous evaluation, we employ an LLM-based judge to assess semantic correctness.

---

> > > ### Author Response · Authors · 2025-11-21
> > >
> > > >**W4: The coverage of the evaluated models is quite restricted**
> > >
> > > We respectfully clarify that our evaluation is not restricted in scope.
> > >
> > > For end-to-end SpeechLMs, we evaluated the three strong capable models that were openly accessible and demonstrated speech-in/text-out instruction following:
> > > - KimiAudio-7B
> > > - Qwen2.5-Omni-7B
> > > - Audio-Flamingo-3
> > >
> > > These systems represented the state of the art (SOTA) in openly available SpeechLM architectures.
> > >
> > > For ASR-LLM pipelines, we evaluated:
> > > - SOTA ASR system Whisper-v3 ASR combined with strong reasoning LLMs across different parameter sizes: Qwen3-8B, Gemma3-27B and Llama3.3-70B
> > >
> > > These pipelines reflect the used real-world deployment patterns and remain competitive baselines for speech-agent tasks. We have further added experiments with an Indic ASR (IndicConformer) in the ASR+LLM pipeline (Results updated in Table 6 in Appendix).
> > >
> > > Overall, the evaluated models span the strongest open SpeechLMs available at the time of submission, as well as robust ASR-LLM pipelines. Our focus was on depth, ensuring rigorous, fine-grained evaluation across all task categories along with meaningful ablation studies. We intend to evaluate newer and proprietary SpeechLMs in the next version of the benchmark to broaden model coverage.
> > >
> > > >**W5**: No confidence intervals or significance tests for the results.
> > >
> > > We acknowledge the reviewer’s feedback on the importance of statistical significance testing. To strengthen the empirical reliability of our findings, we now conduct significance testing on the two primary claims in the paper and the results are tabulated in the Appendix E.2 ( Table 10,11,12,13).
> > >
> > > **1. ASR-LLM pipelines outperform SpeechLMs in most settings.**
> > >
> > > We carried out **McNemar’s test** for all combinations of ASR-LLM and SpeechLM pairs and tabulated the p-values in Tables 10, 11, and 12 in the Appendix. We summarise the results below.
> > >
> > > - Across all three evaluation categories (Single Tool Calling, Single Tool Calling with Retrieval, and Parallel Tool Calling), ASR-LLM systems generally perform better than SpeechLMs.
> > > - The only consistent exception is **KimiAudio-7B**, whose differences against ASR-LLMs are **not always statistically significant, particularly for Qwen2.5-Omni 7B**. However, Whisperv3-Llama70B shows statistically significant differences for Single Tool Calling with Retrieval and Parallel Tool Calling, but not for Single Tool Calling. Whisperv3-Gemma3 27B, on the other hand, shows statistically significant results only for Single Tool Calling with Retrieval.
> > >
> > > **2. KimiAudio-7B significantly outperforms all other SpeechLMs.**
> > >
> > > McNemar tests between SpeechLMs show that KimiAudio-7B consistently outperforms all other SpeechLMs with **significance (p < 0.0001)** across categories and languages.
> > >
> > > **3. Confidence intervals confirm reliability.**
> > >
> > > We also report 95% confidence intervals (CIs) for Parameter Filling (PF)  (Table 13 in Appendix E.2) for KimiAudio across all languages and following three categories: Single Tool Calling, SinTC with Retrieval, and Parallel Tool Calling.
> > >
> > > The CI widths remain relatively compact for every language across the 3 categories (e.g., Bengali: 0.26-0.42, English: 0.61-0.76, Hindi: 0.54-0.71 for Single Tool Calling), indicating that the evaluation sets are sufficiently large to yield reliable, reproducible estimates.

---

> > > > ### Author Response · Authors · 2025-11-21
> > > >
> > > > >**Q2:** Can you quantify how much of the Indic gap remains after substituting better ASR models?
> > > >
> > > > We thank the reviewer for this insightful question. To directly quantify how much of the Indic performance gap arises from ASR errors, we conduct an additional experiment using a recently released *strong Indic ASR model*: **AI4Bharat IndicConformer-600M Multilingual**. We pair this ASR with the same three LLMs used in our main evaluation: Qwen3 8B, Gemma3 27B, and Llama3 70B, and evaluate on the **non-Hindi Indic subset**.
> > > >
> > > > To isolate the effect of ASR quality, we compare three conditions:
> > > > - Ground-truth transcripts + LLM
> > > > - Whisperv3 ASR + LLM
> > > > - IndicConformer ASR + LLM
> > > >
> > > > For all ASR-LLM pipelines, we report the performance drop (Δ) when compared against the ground truth transcript-LLM pipelines. **Table 6 in Appendix D includes the updated comparison.**
> > > >
> > > > **Key Findings:**
> > > > - **IndicConformer substantially reduces degradation:** Using IndicConformer significantly reduces the degradation relative to Whisper. For instance, the performance drop of  30–38 points for Whisperv3 Gemma3 27 model compared to the transcript  shrinks to 10–16 points for IndicConformer Gemma3 27B. Similar patterns are seen for Qwen3 8B and Llama3 70B.
> > > >
> > > > - **Residual gap still remains:** Even with IndicConformer model, PF still lags ground-truth-transcript performance by **10–25 points**, showing that ASR errors, though greatly reduced, continue to propagate into downstream tool-argument prediction.
> > > >
> > > > - **Performance drop in PF increases with task complexity:**  Parallel tool calling sees the highest PF score degradation with Whisper (30–36 points). Indic ASR cuts this substantially (to 23–25 points), but its still higher than the other categories.
> > > >
> > > > - **Larger LLMs benefit more from better ASR:** Gemma3 27B and Llama3 70B recover up to **20–25** points in PF with IndicConformer, but still fall short of transcript-level performance.
> > > >
> > > >
> > > > **Overall Insight:** Using a stronger Indic ASR meaningfully reduces the performance deficit observed with Whisper often recovering **over half of the lost PF accuracy** across non-Hindi Indic languages. However, a noticeable gap remains relative to ground-truth transcripts, even with high-quality Indic ASR. This demonstrates that ASR transcription errors are a major bottleneck for Indic speech inputs. We have **updated Table 6 in Appendix to include the comparison with Indic ASR and have also updated Section 4.3 to include the analysis**. We thank the reviewer for making our evaluation stronger through this experiment.
> > > >
> > > > >**Q3:** Can you provide details in the one-shot settings?
> > > >
> > > > All evaluations in VoiceAgentBench are conducted in a one-shot setting to ensure consistent schema adherence and fair comparison. Each task category includes a category-specific system instruction with a single illustrative example that specifies only the Python tool-calling format without providing task-specific hints. This prevents models from drifting into free-form responses and ensures deterministic parsing. The full set of instructions and one-shot examples is provided in **Appendix J.1.**
> > > >
> > > > We additionally include an ablation (**Table 8, Appendix**) comparing one-shot vs. zero-shot prompting for KimiAudio-7B. Removing the one-shot example leads to large drops in TCS and PF accuracy across Single Tool, SinTC, and Parallel Tool tasks, confirming that one-shot prompting is necessary for stable and valid tool-call generation.

---

> > > > > ### Author Response · Authors · 2025-11-26
> > > > >
> > > > > Dear Reviewer,
> > > > >
> > > > > Thank you for your thoughtful insights. Your feedback helped us significantly strengthen the work, especially through the inclusion of:
> > > > > - human validation study for LLM-as-a-judge
> > > > > - pilot study comparing different TTS systems
> > > > > - statistical tests for our main results
> > > > > - extended ablation by incorporating a Indic ASR into the ASR–LLM pipeline.
> > > > >
> > > > > We hope the updates comprehensively address all of your comments.
> > > > >
> > > > > Thanks!

---

> ### Author Response · Authors · 2025-11-28
>
> Dear Reviewer,
>
> As the discussion period is nearing its conclusion, we want to ensure that all of your concerns have been fully addressed. We have conducted additional analyses and experiments directly aligned with your earlier comments. We look forward to your response.

---

### Official Review · Reviewer_xvuT · 2025-10-31

**Soundness:** 3
**Presentation:** 3
**Contribution:** 2
**Rating:** 4
**Confidence:** 3

**Summary:**

This paper introduces VoiceAgentBench, a novel and comprehensive benchmark designed to systematically evaluate the agentic capabilities of Speech Language Models (SpeechLMs) in realistic spoken interaction scenarios. Unlike existing benchmarks that focus on isolated tasks such as transcription or question answering, VoiceAgentBench assesses a wide range of agentic behaviors, including single and multi-tool invocation, multi-turn dialogue, and safety/adversarial robustness. The benchmark consists of over 5,500 synthetic spoken queries generated via TTS, covering English, Hindi, and five other Indic languages, with a significant portion contextualized in Indian cultural scenarios. The authors propose a diversity-based sampling method for TTS voice conversion to ensure coverage across accents and speaker characteristics, and they define a layered evaluation framework to measure tool selection, structural consistency, parameter filling, and refusal of unsafe requests. Overall, the work fills a critical gap in the evaluation of SpeechLMs for real-world voice assistant applications.

**Strengths:**

1. The proposed evaluation framework is practical and effectively measures agentic capabilities of SpeechLMs in realistic scenarios.

2. The experiments are thorough, covering multiple languages, diverse tasks, and adversarial robustness, with clear and convincing results.

3. The paper is well-structured and clearly written, making the methodology and findings easy to understand.

**Weaknesses:**

1. As a benchmark, the evaluation in this paper is not sufficiently comprehensive; it mainly focuses on agent scenarios in the Indian context, which limits the generality of the assessment.

2. The paper’s originality is somewhat lacking. In the benchmark design, the authors do not clearly explain the fundamental differences between speech agents benchmark and text agents benchmark, nor the necessity of a dedicated speech agent benchmark. The three speech scenarios described seem to be subsets of text agent scenarios. Regarding the claimed innovation of diversity-based TTS generation, it is also unclear whether this approach is necessary, or if using a rich set of audio resources with a zero-shot TTS strategy could achieve similar results.

3. The presentation of the paper could be improved. For the many audio-related cases, a demo page may be a better choice than displaying cases across multiple pages.

**Questions:**

Could the authors clarify the fundamental differences between the proposed speech agent benchmark and existing text agent benchmarks where queries are simply converted to speech using TTS? Specifically, what unique challenges or evaluation aspects are addressed by your benchmark that cannot be captured by applying TTS to text-based agent benchmarks?

---

> ### Author Response · Authors · 2025-11-21
>
> We thank the reviewer for the thoughtful and constructive assessment of our work, and appreciate the recognition of its practical contributions. We further welcome the reviewer’s careful consideration of our methodological choices. We respond to each concern below.
>
> >**W1:** The evaluation in this paper is not sufficiently comprehensive; it mainly focuses on agent scenarios in the Indian context.
>
> We respectfully argue that our proposed VoiceAgentBench benchmark comprehensively covers agent scenarios beyond Indian contexts. English constitutes the single largest portion and is deliberately balanced to create a general and culturally inclusive benchmark across Western and Indian context as well as Indian languages. The exact distribution just for English is as follows:
>
> - English – Western context (Source-native): **1,411 samples (23.01%)**
> - English – Indian context: 965 samples (15.74%)
> - English (Total): **2376 samples (38.75%)**
>
> The remaining 61% of the benchmark is distributed across **six Indic languages**, with Hindi comprising 15.09% and each of the others averaging 9.23%. Thus, the benchmark offers a *balanced multilingual distribution* (Figure 4 in Appendix A) ensuring enough agentic scenarios across English and Indic languages along with Western and Indian Context.
>
> A central goal of VoiceAgentBench is to evaluate **generalization beyond high-resource English**. Including six Indic languages enables evaluation in low-resource settings and reflects realistic deployment scenarios in multilingual regions. The inclusion of Indian contexts does not limit generalizability; rather, it expands it by systematically examining cross-lingual robustness and cultural grounding.
>
> We also acknowledge that some presentation choices may have unintentionally contributed to the perception of disproportionate India-centric focus. Hence, we added Figure 3 in the Appendix A to show English examples and referred to it in the main content. In summary, VoiceAgentBench is intentionally designed to be broadly applicable:
>
> - English remains the largest category (38.75%).
> - The dataset includes both Western and Indian agentic scenarios in English.
> - The multilingual coverage provides necessary evaluation for real-world agentic systems.
> - The benchmark aims to expand beyond narrow monolingual scope by incorporating multilingual and culturally grounded queries.

---

> > ### Author Response · Authors · 2025-11-21
> >
> > > **W2a & Q:** Fundamental differences between speech agents benchmark and text agents benchmark, nor the necessity of a dedicated speech agent benchmark.
> >
> > We appreciate the reviewer’s concern. While it may appear that speech-agent evaluation could be approximated by simply applying TTS to existing text-agent benchmarks, this assumption does not hold in practice.
> >
> > Our benchmark introduces several fundamental innovations, both in task design and in data construction that are neither captured by text benchmarks nor by naively converting text datasets to speech.
> >
> > **1. No existing text benchmark covers multilingual Indic agentic scenarios or culturally grounded tasks:** To the best of our knowledge, there is no text-based agentic benchmark that provides Indic multilingual tasks. We also provide culturally grounded, and domain-diverse agentic queries, thus contributing new agentic evaluation tasks even in the text modality.
> >
> > **2. We introduce custom agent tools supporting sequential multi-step workflows, absent from prior text benchmarks:** Existing text agent benchmarks predominantly test single or parallel function calling. We introduce 21 new custom tools that support multi-step, stateful, sequential workflows inspired by real-world voice assistants (cab booking, food ordering, payments, etc.). These capabilities are not generally covered in text benchmarks. Evaluating sequential tool workflows is one of the core contributions of our work.
> >
> > **3. Spoken queries are not equivalent to written queries; we design queries explicitly for speech:** Spoken interactions require different data design choices than text: shorter and more conversational user queries. We carefully design and choose queries that are speech-native, reflecting real usage.
> >
> > **4. Our speech generation is non-trivial and cannot be produced by straightforward TTS conversion.** Our data construction pipeline:
> > - use language-specific TTS engines with diverse speakers, based on Mean Opinion Score (MOS) analysis (Appendix N);
> > - employ diversity sampling strategy to induce realistic variability;
> > - ensure high-quality translations by filtering for script mixing and unnatural repetitions;
> > - align speech durations with the natural cadence expected in real-world voice assistants.
> >
> > These details are essential for a realistic speech-native benchmark and cannot be achieved by a naive TTS conversion.
> >
> > **5. We provide a unified framework covering a taxonomy of six evaluation categories**, not collectively supported in any prior speech or even text based agentic benchmarks.
> >
> >  VoiceAgentBench, simultaneously evaluates:
> > - Single Tool Calling
> > - Single Tool Calling with Retrieval
> > - Parallel Tool Calling
> > - Sequential Tool Calling
> > - Multi-turn Conversational Tool Use
> > - Safety & Refusal Behavior
> >
> > **In summary**, VoiceAgentBench offers clear originality in task design, multilingual coverage, tool ecosystems, speech-native construction, and evaluation methodology. Our contributions underscore the need for a dedicated speech-agent benchmark and distinguish our work from prior text-based efforts.
> >
> > >**W2b**: Whether diversity-based TTS generation is necessary, or if using a rich set of audio resources with a zero-shot TTS strategy could achieve similar results.
> >
> > We thank the reviewer for raising this important point. In the initial submission, we evaluated three principled diversity-driven sampling strategies: Density-Based Sampling, Farthest-Point Sampling (FPS), and Determinantal Point Processes (DPP) to construct a representative synthetic audio dataset. Based on the suggestion, we additionally include a random sampling baseline drawn from the same underlying pool of speakers simulating the zero-shot TTS setting.
> >
> > As shown in the updated t-SNE plots (**Figure 5 in Appendix**) , **random sampling yields the lowest** *mean distance to the nearest selected point* among all methods. This indicates that zero-shot TTS audio generation would fail to create a more diverse distribution of user acoustic variability. In contrast, FPS consistently produces more diverse samples. We have updated **Appendix C and Sec 3.2.4** to reflect addition of random sampling as part of our comparison.
> >
> > > **W3:** A demo page may be a better choice than displaying cases across multiple pages.
> >
> > We agree with the reviewer that an interactive demo is a more convenient way to browse the benchmark than static examples listed across multiple appendices. To address this, we have prepared a lightweight demo page that provides category-wise speech examples from VoiceAgentBench: https://anonymous.4open.science/r/vab-2833/Examples.md. The demo also contains some failure cases of  SpeechLMs: https://anonymous.4open.science/r/vab-2833/Failure_Cases.md
> >
> > Furthermore, when we open-source the full benchmark, we will include a complete interactive demo to make exploration of multilingual queries, tool workflows, and speech diversity even more accessible.

---

> > > ### Author Response · Authors · 2025-11-26
> > >
> > > Dear Reviewer,
> > >
> > > Thank you for your thoughtful insights. Your feedback helped strengthen our work, both by reinforcing our claim on diversity sampling through a comparison with a random-sampling baseline, and by improving clarity through additional explanations on English-context distribution and a demo.
> > >
> > > We hope this response fully addresses all of your comments.
> > >
> > > Thanks!

---

> > > > ### Author Response · Authors · 2025-11-28
> > > >
> > > > Dear Reviewer,
> > > >
> > > > With the discussion phase wrapping up soon, we wanted to check in and confirm that our additional experiments and clarifications have satisfactorily resolved your earlier concerns. We truly appreciate your time and engagement, and we look forward to hearing back from you.

---

### Official Review · Reviewer_nz5n · 2025-11-01

**Soundness:** 2
**Presentation:** 2
**Contribution:** 2
**Rating:** 4
**Confidence:** 5

**Summary:**

This work introduces Voiceagentbench, a new benchmark specifically designed to evaluate the agent-based (speech-to-text) functionalities of Speech LLMs. The benchmark comprises over 5,000 synthetic spoken queries, including dialogues grounded in Indian contexts, and is structured to cover single-tool invocations, multi-tool workflows, multi-turn interactions, and safety evaluations. The dataset construction follows a conventional pipeline: text prompts are sourced from existing text-based LLM agent benchmarks and then converted to speech via TTS synthesis. The authors evaluate several prominent models, including qwen2.5-omni, kimi-audio, and audioFalmingo3, alongside various cascaded speech-to-text LLM systems. The results indicate that, with the exception of kimi-audio, the other end-to-end models (qwen2.5-omni, audioFalmingo3) exhibit significant performance degradation on this benchmark.

**Strengths:**

1. Voiceagentbench represents a timely and pioneering effort, being one of the first benchmarks dedicated to evaluating the agent capabilities of speech-based LLMs.

2. Although the methodology for dataset creation is not highly innovative, the resulting dataset, pipeline, and evaluation framework provide a contribution that will be useful to the research community.

**Weaknesses:**

1. The dataset is heavily skewed towards Indian contexts, in both its textual content and synthesized audio. This poses a significant domain mismatch for mainstream models, which are predominantly trained on general English data. This specialization limits the benchmark's broader applicability and generalizability.
2.  Authors need to clarify the relationship between their work and existing research on voice agents. A more thorough discussion is required to situate this benchmark relative to recent advancements, such as[1][2][3][4]:
3. The benchmark's design appears to primarily target cascaded speech-to-text LLMs, rather than being optimized for evaluating true end-to-end speech agent models.
4. The exclusive use of synthetic data introduces a potential gap between the benchmark's evaluation scenarios and the complexities of real-world, spontaneous human-machine conversation.
5. Many of the conclusions drawn from the experiments offer limited insight. For instance, the poor performance of qwen2.5-omni (often scoring 0) is likely attributable to a simple lack of agent-specific training data, rather than a deeper flaw in the model's architecture. Such conclusions do not foster deeper investigation into the core challenges of speech agents.
6. The overall pipeline—dataset creation, model selection, and evaluation—is conventional, albeit representing a non-trivial amount of work. The benchmark overlooks several critical dimensions of agent performance that warrant further exploration, such as: The latency introduced by agent function calls. The potential degradation of the base model's core abilities when agent functionalities are integrated. The evaluation of complex, multi-agent collaborative scenarios.




[1]. Process-Supervised Reinforcement Learning for Interactive Multimodal Tool-Use Agents

[2]. Stream RAG: Instant and Accurate Spoken Dialogue Systems with Streaming Tool Usage

[3]. GPT-RealTime

[4]. Step-Audio 2 Technical Report

**Questions:**

It would be highly beneficial for the authors to include a more comprehensive set of examples or case studies from the dataset. This would provide greater transparency and a clearer understanding of the benchmark's tasks and data quality.

---

> ### Author Response · Authors · 2025-11-21
>
> We thank the reviewer for the thoughtful and constructive assessment of our work. We appreciate the recognition that VoiceAgentBench represents one of the first dedicated benchmarks for evaluating speech-based agentic capabilities, and that our work offer meaningful value to the research community. Below, we address the reviewer’s concerns in detail.
>
> >**W1:** The dataset is heavily skewed towards Indian contexts
>
> We respectfully argue that our proposed VoiceAgentBench benchmark is not skewed towards Indian contexts. In fact, English constitutes the single largest portion and is deliberately balanced to create a general and culturally inclusive benchmark across Western and Indian context as well as Indian languages. The exact distribution just for English is as follows:
>
> - English – Western context (Source-native): **1,411 samples (23.01%)**
> - English – Indian context: 965 samples (15.74%)
> - English (Total): **2376 samples (38.75%)**
>
> The remaining 61% of the benchmark is distributed across **six Indic languages**, with Hindi comprising 15.09% and each of the others averaging 9.23%. Thus, the benchmark offers a *balanced multilingual distribution* (Figure 4 in Appendix A) ensuring enough coverage across English and Indic languages along with Western and Indian Context.
>
> A central goal of VoiceAgentBench is to evaluate **generalization beyond high-resource English**. Including six Indic languages enables evaluation in low-resource settings and reflects realistic deployment scenarios in multilingual regions. The inclusion of Indian contexts does not limit generalizability; rather, it expands it by systematically examining cross-lingual robustness and cultural grounding.
>
> We also acknowledge that some presentation choices may have unintentionally contributed to the perception of disproportionate India-centric focus. Hence, we added Figure 3 in the Appendix A to show English examples and referred to it in the main content. In summary, VoiceAgentBench is intentionally designed to be broadly applicable:
>
> - English remains the largest category (38.75%).
> - The dataset includes both Western and Indian context in English.
> - The multilingual coverage provides necessary evaluation for real-world agentic systems.
> - The benchmark aims to expand beyond narrow monolingual scope by incorporating multilingual and culturally grounded queries.
>
> >**W2:** Authors need to clarify the relationship between their work and existing research on voice agents.
>
> We thank the reviewer for highlighting recent voice-agent works. We note, however, that these papers were released very close to or even after our submission deadline. While they present valuable advances in voice assistants, their focus is primarily on training and architectural improvements (e.g., RL-based training, low-latency streaming) with a very limited evaluation of speech-based tool use.
>
> In contrast, VoiceAgentBench provides the first comprehensive and culturally inclusive benchmark for speech-native agents, offering fine-grained, multilingual, tool-centric evaluation across single/parallel/sequential tool use, multi-turn dialogue, safety refusal, and culturally grounded Indic scenarios which are not covered in prior as well as contemporary speech-agent research.
>
> We have clarified this positioning and included all these suggested recent works in the revised related-work section (Appendix B).
>
> > **W3:** The benchmark's design appears to primarily target cascaded speech-to-text LLMs, rather than being optimized for evaluating true end-to-end speech agent models.
>
> While this is an interesting direction, our benchmark is intentionally designed to evaluate explicit **speech-in agentic tool use** which is why the evaluation protocol centers on tool-calling accuracy rather than penalizing or rewarding TTS capabilities. While recent end-to-end speech-in/speech-out systems are an exciting direction, modeling expressive spoken responses is orthogonal to the core goal of our work: **measuring grounded reasoning, tool selection, schema adherence, and parameter filling** from spoken queries. Extending the benchmark to also assess speech-out generation is a valuable future direction, but it is outside the scope of the current submission, which focuses specifically on **agentic decision-making from speech.**

---

> > ### Author Response · Authors · 2025-11-21
> >
> > >**W4:** The exclusive use of synthetic speech introduces a potential gap between the benchmark and real-world spontaneous human–machine conversation.
> >
> > We understand the reviewer’s concern regarding synthetic speech. However, generating large-scale real spoken instructions is recognized as challenging and costly. As noted in prior work such as VoiceBench [1], collecting real human speech at scale is often impractical:
> >
> > *“Due to the high cost of generating real spoken instructions, relying solely on actual speech to evaluate voice assistants comprehensively is challenging. As a result, after preparing the text-based instructions, we convert them into speech using text-to-speech (TTS) models.”*
> >
> > Other work such as IndicSynth [2] also highlights how recent advances in TTS and voice-conversion models now enable the generation of high-quality synthetic speech that closely emulates natural human voices. Further, to ensure meaningful variability in our setup, we perform diversity-based speaker sampling that spans accents, ages, genders, prosodic patterns, and speaking styles. Additionally, we conducted a pilot study (Appendix N) to identify language-specific TTS pipelines, which formed the basis for the selection of ElevenLabs for English, and Krutrim-TTS for Indic languages, ensuring that generated audio preserves native phonetic and prosodic characteristics.
> >
> > That said, we agree on the importance of real/noisy speech. As also acknowledged in Section 5 (Limitations Line 499), the future version of VAB will include:
> > - Real human speech collected across 7 languages
> > - Additive environmental noise and disfluency augmentation
> > - Accent-balanced natural speech following the same diversity metrics
> >
> > [1] VoiceBench: Benchmarking LLM-Based Voice Assistants
> >
> > [2] IndicSynth: A Large-Scale Multilingual Synthetic Speech Dataset for Low-Resource Indian Languages (ACL 2025)
> >
> > >**W5:** Simple lack of agent-specific training data, rather than a deeper flaw in the model's architecture
> >
> > We agree that the weaker performance of models such as Qwen2.5-Omni is primarily due to limited exposure to structured agentic supervision rather than shortcomings of their architecture. Our goal in the paper is not to imply architectural inadequacy, but instead to highlight a clear gap in agent-specific capabilities for current state of application in SpeechLMs. This is consistent with our discussion in the paper, where we note that end-to-end SpeechLMs owing to their low TTFT and ability to integrate agentic context during speech decoding stand to benefit substantially from additional training on tool-use oriented datasets. We view our benchmark as a diagnostic tool that reveals where such supervision is lacking, and as a resource intended to motivate further fine-tuning with RL, or SFT efforts for building stronger speech-native agent models.
> >
> > >**W6:** The benchmark overlooks several critical dimensions of agent performance that warrant further exploration.
> >
> > We acknowledge observations regarding latency, potential degradation of core model abilities, and multi-agent evaluation. These are indeed important considerations for building practical speech-agent systems. However, they fall outside the scope of this first iteration of VoiceAgentBench, whose primary goal is to establish a foundational, reproducible evaluation suite for agentic competence from speech focusing on correct tool selection, argument grounding, multilingual robustness, and safety refusal behavior.
> >
> > Latency, while crucial in deployment and final user experience, does not meaningfully inform comparisons of models’ task performance, however we compare the general latency in terms of TTFT for ASR-LLM and SpeechLMs in Appendix F.
> > Similarly, our benchmark does not evaluate potential degradation of core model abilities because **we do not fine-tune or modify any SpeechLMs**; all models are assessed strictly in their released form with no task-specific adaptation. Investigating how agent-oriented supervision or fine-tuning influences underlying model capabilities is an important direction, but it falls outside the scope of this work.
> >
> > Finally, while multi-agent collaboration is an exciting frontier, robust evaluation of multi-agent systems first requires reliable single-agent competence. VoiceAgentBench is, to our knowledge, the first benchmark to rigorously evaluate SpeechLMs across a fine-grained taxonomy of single-tool, multi-tool (parallel and sequential), dialogue-based, and realistic multi-step workflows. Our findings show that current models still struggle with these single-agent tasks, especially in Indic multilingual settings, suggesting that multi-agent coordination is premature at this stage. We view multi-agent scenarios as a natural next step and plan to extend VoiceAgentBench in that direction as models mature.

---

> > > ### Author Response · Authors · 2025-11-21
> > >
> > > >**Q:** Include a more comprehensive set of examples or case studies from the dataset.
> > >
> > > We thank the reviewer for the suggestion. We have expanded the presentation of examples in multiple parts of the appendix. **Appendix G** illustrates how our evaluation framework identifies and categorizes different model failures (Incorrect Tool Selection, Schema Violations, and Parameter Filling Errors). **Appendix H** provides clear illustrations of the *custom agent tools* we designed to enable sequential, multi-step workflows, capabilities not covered by existing benchmarks. **Appendix I** presents diverse, category-wise examples sampled across all task types in VoiceAgentBench, and **Appendix K** further includes multilingual examples spanning every supported language.
> > >
> > > In addition, we have prepared a **lightweight demo** showcasing representative speech samples directly from VoiceAgentBench: https://anonymous.4open.science/r/vab-2833/Examples.md
> > >
> > > The demo also contains some cases of failure in SpeechLMs: https://anonymous.4open.science/r/vab-2833/Failure_Cases.md

---

> > > > ### Author Response · Authors · 2025-11-26
> > > >
> > > > Dear Reviewer,
> > > >
> > > > Thank you for your valuable insights and questions. Your feedback prompted us to clarify the English context distribution in VoiceAgentBench and prepare a demo that more comprehensively presents samples and failure cases from the benchmark.
> > > >
> > > > We would like to confirm that all your comments have been fully addressed.
> > > >
> > > > Thanks!

---

> > > > > ### Comment · Reviewer_nz5n · 2025-11-27
> > > > >
> > > > > Thank you very much for the authors' response, especially for providing the supplementary demo examples. I do acknowledge the partial contributions (as noted in the "Strength" section) of VoiceAgentBench.
> > > > >
> > > > > However, after carefully reading the authors' rebuttal, I still maintain all of the points I raised in the "Weakness" section. These include, but are not limited to: the need for further exploration of factors such as latency; the necessity of validating the logic of tool invocation on speech-to-speech models; the requirement for a comparative analysis of the data portion of related works (both pre- and post-ICLR); the limiting of utility by focusing heavily on the Indian accent/scenario; and the need for validation in real-world scenarios.
> > > > >
> > > > > This work does not meet my personal standards for acceptance at ICLR, and therefore I choose to maintain my current score. Nevertheless, I recognize this as a valuable contribution with high timeliness and suggest that the authors consider publishing it in other venues.
> > > > >
> > > > > Best wishes.

---

> > > > > > ### Author Response · Authors · 2025-11-28
> > > > > >
> > > > > > We thank the reviewer for the reply and the time. We would like to provide further clarification on the mentioned concerns:
> > > > > >
> > > > > > > **Benchmarking Latency**
> > > > > >
> > > > > > We appreciate the reviewer’s concern about latency. However, latency depends on hardware (H100, A100s), inference engines (like vLLM), and model-specific optimization support. We do provide a Time taken to first token (TTFT) comparison between speechLM and ASR-LLM in Appendix F:
> > > > > >
> > > > > > *“When measured with a set of 100 queries of average duration 3.5 seconds, Qwen2.5-Omni 7B achieves a 90th percentile (p90) TTFT of approximately 40 ms on a single H100 GPU, whereas a pipeline combining Whisper-large-v3 with Qwen3 8B exhibits a p90 TTFT of around 800 ms under the same hardware conditions.”*
> > > > > >
> > > > > > In our evaluation, we ensured that each model was evaluated under its best possible inference configuration. For instance, the ASR-LLM pipeline along with Qwen2.5-Omni benefits from compatibility with high-performance inference stacks such as **vLLM**. This yields significantly faster LLM decoding. In contrast, several SpeechLMs do not currently support vLLM or comparable optimized runtimes, and forcing them into a shared but suboptimal inference setup would have resulted in an unfair comparison for latency.
> > > > > >
> > > > > > As SpeechLMs gain support for optimized inference engines, we anticipate more equitable latency comparisons to become feasible.
> > > > > >
> > > > > > > **Necessity of validating the logic of tool invocation on speech-to-speech models**
> > > > > >
> > > > > > Tool invocation is fundamentally a text-based operation: even when the model supports spoken input and output, the actual function call is issued in text, executed externally, and the resulting information is then fed back to the model to produce a spoken response. **Our benchmark is designed to evaluate this core text-level tool-calling capability, which is shared across both cascaded speech-to-text and end-to-end speech-to-speech architectures**.
> > > > > >
> > > > > > Models such as KimiAudio and Qwen2.5-Omni, which support speech-to-speech interaction, are already included in our evaluation. Since the focus of our benchmark is to measure whether the model selects the correct tool, formats the call properly, and fills parameters accurately, it intentionally isolates tool-invocation ability from the separate problem of generating a coherent spoken response after the tool call.
> > > > > >
> > > > > > This mirrors standard practice in the ASR and TTS communities, where components are evaluated separately: ASR using WER/CER, TTS using MOS or speaker similarity, before being integrated into full speech systems. Similarly, **our benchmark isolates and evaluates the component of speech-to-speech agents responsible for reasoning and tool invocation**. Our results show that even SOTA models struggle with this core capability, underscoring the importance of evaluating tool-calling logic independently of downstream speech generation.
> > > > > >
> > > > > > > **Comparative analysis of the data portion of related works (both pre- and post-ICLR)**
> > > > > >
> > > > > > We have cited these concurrent works in our related-work section. [1] converts the CRAG benchmark [2] into speech using an in-house TTS system. Similarly [3] performs evaluation by converting Tau-Bench [4] into spoken format through SeedTTS [5]. While methodologically aligned with our use of TTS, their evaluations are relatively narrow, focusing primarily on single-turn tasks without broader agentic behaviors. In contrast, VoiceAgentBench provides a substantially more comprehensive foundation: multi-turn dialogue, sequential and parallel tool calling, safety evaluation, multilingual and culturally grounded queries, and diversity-aware TTS generation. As such, it enables evaluating not only these existing methodologies but also future speech-agent systems under richer and more realistic conditions.
> > > > > >
> > > > > > [1] Stream RAG: Instant and Accurate Spoken Dialogue Systems with Streaming Tool Usage
> > > > > >
> > > > > > [2] Corrective Retrieval Augmented Generation
> > > > > >
> > > > > > [3] Process-Supervised Reinforcement Learning for Interactive Multimodal Tool-Use Agents
> > > > > >
> > > > > > [4] τ-bench: A Benchmark for Tool-Agent-User Interaction in Real-World Domains
> > > > > >
> > > > > > [5] Seed-tts: A family of high-quality versatile speech generation models

---

> > > > > > > ### Author Response · Authors · 2025-11-28
> > > > > > >
> > > > > > > >**Need for validation in real-world scenarios**
> > > > > > >
> > > > > > > We want to stress again that the inclusion of custom agents for sequential tool calling was intentionally motivated by real-world voice assistant use cases. **Tasks such as cab booking, food ordering, and financial transactions reflect the types of multi-step interactions deployed in commercial agentic systems**. These domains inherently require planning, grounding, and accurate parameter extraction capabilities that voice assistants must demonstrate for practical deployment. Our results show that both SpeechLMs and ASR-LLM pipelines struggle in these scenarios.
> > > > > > >
> > > > > > > Similarly, our layered evaluation metric: covering *Tool Selection, Tool Call Structure, and Parameter Filling*, was designed to reflect the competencies needed for reliable voice-agent operation. Each layer corresponds to a necessary component of end-to-end task completion in actual agent workflows. In addition, VoiceAgentBench incorporates **diversity in TTS** to capture the heterogeneity encountered in real usage, along with **safety-sensitive tasks**, which are essential for assessing whether real-world agents avoid harmful tool invocation.
> > > > > > >
> > > > > > > Together, these choices ensure the VoiceAgentBench faithfully represents the challenges and risks of real-world voice-agent deployments.
> > > > > > >
> > > > > > > We believe our contributions are substantively novel, and that VoiceAgentBench provides a foundational, scalable framework for evaluation of current and future voice assistants. We are also open to suggestions on any additional experiments or improvements in presentation that would help strengthen the manuscript.

---

### Official Review · Reviewer_Tfdb · 2025-11-01

**Soundness:** 3
**Presentation:** 3
**Contribution:** 3
**Rating:** 6
**Confidence:** 4

**Summary:**

This paper introduces VoiceAgentBench (VAB), the first benchmark designed to evaluate agentic speech-based systems---i.e., voice assistants that can reason, call tools, and handle multi-step workflows. The benchmark includes 5,500+ synthetic spoken queries in 7 languages (English + 6 Indic), covering:

- Single-tool and multi-tool usage
- Sequential dependent workflows
- Multi-turn interactions
- Safety and adversarial robustness

Six models (SpeechLMs/ASR to LLM pipelines) are evaluated. Results highlight significant weaknesses in multilingual performance, tool reasoning, and safety alignment.

**Strengths:**

1. Novel and timely: first benchmark for speech-based agentic reasoning and tool orchestration.
2. Comprehensive task design: Includes sequential tool use, multi-tool planning, multi-turn dialogue, and adversarial safety tasks.
3. Multilingual and culturally grounded: covers 7 languages, with 30% Indian-context queries, which has been rare in previous work.
4. Acoustic diversity: uses Farthest Point Sampling (FPS) over TTS speaker embeddings to maximize voice variation.
5. Granular evaluation metrics. Tool Selection (TS), Tool Call Structure (TCS), Parameter Filling (PF), and Refusal Rate (RR) allow fine-grained diagnostics.
6. Extensive experiments and ablations, including error propagation from ASR, prompting strategies, and safety prompting.

**Weaknesses:**

1. All speech is synthetic (TTS), which limits realism and excludes natural speech variability, noise, and disfluencies.
2. Ground truth for sequential tool workflows is generated using GPT-4o-mini rather than human annotation, which may introduce hallucinations or inaccuracies.
3. Evaluation relies on GPT-4o-mini as a judge for parameter accuracy without human agreement studies or reproducibility checks.
4. Multi-turn dialogue is only evaluated in English and entirely missing for Indic languages.
5. Safety evaluation treats refusal as a binary outcome. And also does not seem to assess partial compliance, harmful outputs, or nuanced behavior.
6. No statistical significance testing, confidence intervals, or variance reporting; dataset construction and filtering steps lack transparency.

**Questions:**

- Has any portion of the GPT-generated ground truth been validated by humans?
- What is the agreement between human judges and GPT-4o-mini in evaluation?
- Why were multi-turn tasks excluded for Indic languages, and will they be added?
- Do you plan to include real or noisy speech in future versions of the benchmark?
- Can you provide per-language breakdowns of failure types?

---

> ### Author Response · Authors · 2025-11-21
>
> We thank the reviewer for the thoughtful and constructive feedback. We are encouraged by the recognition of the multilingual and culturally grounded design, acoustic diversity, evaluation depth and the novelty of VoiceAgentBench. We address all the concerns below.
>
>
> > **W1 & Q4:** All speech is synthetic (TTS), which limits realism and excludes natural speech variability, noise, and disfluencies.
>
> We understand the reviewer’s concern here. However, generating large-scale real spoken instructions is both challenging and costly. As noted in prior work such as VoiceBench [1], collecting real human speech at scale is often impractical:
>
> *“Due to the high cost of generating real spoken instructions, relying solely on actual speech to evaluate voice assistants comprehensively is challenging. As a result, after preparing the text-based instructions, we convert them into speech using text-to-speech (TTS) models.”*
>
> Similarly, other works [2] argue that recent high-quality TTS systems produce synthetic speech that serves as a reliable proxy for human voices. Further, to ensure meaningful variability in our setup, we perform diversity-based speaker sampling that spans accents, gender, prosodic patterns, and speaking styles. Additionally, we conducted a pilot study (Appendix N) to identify language-specific TTS pipelines, which formed the basis for the selection of ElevenLabs for English, and Krutrim-TTS for Indic languages, ensuring that generated audio preserves native phonetic and prosodic characteristics.
>
> That said, we agree on the importance of real/noisy speech. As also acknowledged in Section 5 (Line 499), the future version of VAB will include:
> - Real human speech collected across 7 languages
> - Additive environmental noise and disfluency augmentation
> - Accent-balanced natural speech following the same diversity metrics
>
> [1] VoiceBench: Benchmarking LLM-Based Voice Assistants
>
> [2] IndicSynth: A Large-Scale Multilingual Synthetic Speech Dataset for Low-Resource Indian Languages (ACL 2025)
>
>
> > **W2:** Ground truth for sequential tool workflows is generated using GPT-4o-mini rather than human annotation, which may introduce hallucinations or inaccuracies.
>
> We appreciate the raised concerns. Since sequential tool workflows are relatively less in number and are not sourced from pre-existing datasets, each instance was custom-curated and subsequently manually reviewed by authors to ensure correctness, completeness, and logical consistency across chained calls. Apart from this, like all other categories, it went through deterministic Pydantic schema checks for tool call structure.
>
> We have updated Section 3.2.2 (Line 248) to explicitly mention this verification process.
>
> In addition, we have prepared a lightweight public demo showcasing representative speech samples directly from VoiceAgentBench: https://anonymous.4open.science/r/vab-2833/Examples.md
>
> >**W3 & Q2:** Evaluation relies on GPT-4o-mini as a judge for parameter accuracy without human agreement studies or reproducibility checks.
>
> We thank the reviewer for highlighting the importance of validating LLM-judge reliability. To assess the reproducibility and human alignment of our GPT-4o-mini–based evaluation for parameter-filling accuracy, we conducted an additional human agreement study on a representative subset of our benchmark. We find significant agreement between GPT judge and human responses (with average **0.92** pairwise agreement while kappa average of **0.79**).
>
> **Experimental setup:** We sampled ~200 instances across categories and languages and collected ~400 model responses from KimiAudio and Whisper-Llama3.3-70B. Each response was labeled as correct or incorrect by two independent expert annotators (Annotator 1 and Annotator 2). We additionally compute a human-majority label, defined as the consensus between the two annotators for agreement analysis. We evaluate alignment using two complementary metrics where we have substantial agreement:
>
> **1. Pairwise Agreement:**
>
> This measures raw agreement rates between the GPT-4o-mini judge and human annotators.
>
> - GPT-4o-mini vs. Annotator 1: 0.9039
> - GPT-4o-mini vs. Annotator 2: 0.9360
> - GPT-4o-mini vs. Human Majority: 0.9236
> - Annotator 1 vs. Annotator 2: 0.9680
>
> **2. Cohen’s Kappa:**
>
> We additionally report Cohen’s Kappa to assess reliability of GPT-4o-mini as a judge.
>
> - GPT-4o-mini vs. Annotator 1: 0.7488
> - GPT-4o-mini vs. Annotator 2: 0.8310
> - GPT-4o-mini vs. Human Majority: 0.7953
> - Annotator 1 vs. Annotator 2: 0.9141
>
> We have added the details and results of this experiment in the Appendix M and have updated Section 3.3 to mention this.

---

> > ### Author Response · Authors · 2025-11-21
> >
> > >**W4 & Q3:** Multi-turn dialogue is only evaluated in English and entirely missing for Indic languages.
> >
> > Our data-generation pipeline is inherently scalable across languages and naturally supports complex multi-turn conversational structures, making extension to additional languages straightforward. The absence of multi-turn evaluation for Indic languages in the initial submission stemmed primarily from time constraints and our focus on ensuring uniformly high-quality speech generation across all languages. Since submission, we have constructed and evaluated a multi-turn Hindi subset consisting of 374 samples with the following results (also updated in Table 2 in Section 4 of the paper):
> >
> > | Model                    | Tool Selection | Tool Call Structure | Parameter Filling |
> > |--------------------------|----------------|----------------------|--------------------|
> > | Qwen2.5-Omni 7B          | 69.79          | 1.34                 | 1.07               |
> > | KimiAudio 7B             | 73.26          | 67.91                | 28.61              |
> > | Whisperv3-Qwen3 8B       | 93.85          | **91.44**            | 37.70         |
> > | Whisperv3-Gemma3 27B     | 94.39          | 86.36                | 35.83              |
> > | Whisperv3-Llama3 70B     | **97.33**      | 86.10                | **39.30**          |
> >
> > The multi-turn Hindi results are consistent with the broader findings reported in our main results:
> > - Firstly, within the SpeechLM, **KimiAudio 7B again emerges as the strongest model**, achieving 73.26% TS, 67.91% TCS, and 28.61% PF, substantially higher than Qwen2.5-Omni 7B (69.79% TS, 1.34% TCS, 1.07% PF). This mirrors the English subset, where KimiAudio consistently outperformed Qwen2.5-Omni 7B in reasoning-heavy tool-calling tasks.
> >
> > - Secondly, ASR-LLM pipelines continue to outperform all SpeechLMs by a wide margin in the multi-turn setting. Whisper-v3-Llama3 70B and Whisper-v3-Gemma3 27B reach 97.33% / 86.10% / 39.30% and 94.39% / 86.36% / 35.83% (TS / TCS / PF), respectively, far above the SpeechLMs.
> >
> > - Finally, both SpeechLMs and ASR-LLM pipelines exhibit a **drop in Hindi multi-turn performance compared to the English subset** (e.g., Whisper-v3-Llama3 70B decreases from 93.43% → 86.10% TCS, and 61.62% → 39.30% PF). This aligns with our broader observation that multilingual and culturally grounded tasks introduce measurable degradation for both SpeechLMs and ASR-LLM pipelines.
> >
> > Overall, the Hindi multi-turn results reinforce our three central findings: (i) KimiAudio remains the strongest SpeechLM across languages, (ii) ASR-LLM pipelines remain decisively superior in multi-turn settings, and (iii) all models experience noticeable performance drops when shifting from English to Hindi.
> >
> > We could not evaluate the multi-turn setting for AudioFlamingo-3 because the released codebase did not support multi-turn inference. We have updated the statistics of the benchmark (Table 1 in Section 3.1) in our paper to reflect the addition of this subset. In future iterations, we plan to extend this multi-turn evaluation to all remaining Indic languages.
> >
> > >**W5:** Safety evaluation treats refusal as a binary outcome. And also does not seem to assess partial compliance, harmful outputs, or nuanced behavior.
> >
> > Our safety evaluation follows the refusal-rate metric established in AgentHarmBench [1], where the central criterion is whether the model initiates any tool invocation in response to a harmful user query. This criteria was a deliberate design choice since in agentic settings, even partial compliance such as executing only the first step of a harmful workflow constitutes actionable harm. Consequently, safety behavior is operationalized as a binary decision: either the model fully refuses, or it begins executing a harmful request.
> >
> > Regarding nuanced behavior, we conducted two targeted ablations (Section 4.3) in our work:
> > - Impact of Refusal Prompts on Safety Behavior (Appendix, Fig. 6)
> > - Effect of Adversarial Hints on Refusal Rates. (Appendix, Fig. 7)
> >
> > While our primary metric is intentionally binary for safety-critical reasons, our evaluation does capture more nuanced safety dynamics through controlled ablations on prompt structure and adversarial pressure.
> >
> > [1] AgentHarm: A Benchmark for Measuring Harmfulness of LLM Agents [ICLR 2025]

---

> > > ### Author Response · Authors · 2025-11-21
> > >
> > > >**W6a**: No statistical significance testing, confidence intervals, or variance reporting
> > >
> > > We acknowledge the reviewer’s feedback on the importance of statistical significance testing. To strengthen the empirical reliability of our findings, we now conduct significance testing on the two primary claims in the paper and the results are tabulated in the Appendix E.2 ( Table 10,11,12,13).
> > >
> > > **1. ASR-LLM pipelines outperform SpeechLMs in most settings.**
> > >
> > > We carried out **McNemar’s test** for all combinations of ASR-LLM and SpeechLM pairs and tabulated the p-values in Tables 10, 11, and 12 in the Appendix. We summarise the results below.
> > >
> > > - Across all three evaluation categories (Single Tool Calling, Single Tool Calling with Retrieval, and Parallel Tool Calling), ASR-LLM systems generally perform better than SpeechLMs.
> > > - The only consistent exception is **KimiAudio-7B**, whose differences against ASR-LLMs are **not always statistically significant, particularly for Qwen2.5-Omni 7B**. However, Whisperv3-Llama70B shows statistically significant differences for Single Tool Calling with Retrieval and Parallel Tool Calling, but not for Single Tool Calling. Whisperv3-Gemma3 27B, on the other hand, shows statistically significant results only for Single Tool Calling with Retrieval.
> > >
> > > **2. KimiAudio-7B significantly outperforms all other SpeechLMs.**
> > >
> > > McNemar tests between SpeechLMs show that KimiAudio-7B consistently outperforms all other SpeechLMs with **significance (p < 0.0001)** across categories and languages.
> > >
> > > **3. Confidence intervals confirm reliability.**
> > >
> > > We also report 95% confidence intervals (CIs) for Parameter Filling (PF)  (Table 13 in Appendix E.2) for KimiAudio across all languages and following three categories: Single Tool Calling, SinTC with Retrieval, and Parallel Tool Calling.
> > >
> > > The CI widths remain relatively compact for every language across the 3 categories (e.g., Bengali: 0.26-0.42, English: 0.61-0.76, Hindi: 0.54-0.71 for Single Tool Calling), indicating that the evaluation sets are sufficiently large to yield reliable, reproducible estimates.
> > >
> > > >**W6b**: Dataset construction and filtering steps lack transparency.
> > >
> > > We respectfully disagree here, Section 3.2 clearly outlines our data construction pipeline. We summarize the key steps here for clarity.
> > >
> > > VoiceAgentBench is built by assembling tools from established benchmarks (BFCL, API-Bank, AgentHarm) and by designing **21 custom tools** for realistic sequential workflows (cab booking, food ordering, payments).
> > >
> > > We include both **source-native** (English) examples and culturally grounded **Indian-context** queries. The latter are generated using **Gemma3-27B and GPT-4o-mini**. Multilingual examples (Indic languages) are filtered using lightweight quality checks on **language detection, script mixing, and repetition**, removing malformed generations (drawing on prior work [1] filtering checks).
> > >
> > > For speech, we employ a diversity-based TTS pipeline to ensure broad variation in accents, prosody, and gender. We use **ElevenLabs + Coqui-TTS** for English and **Krutrim-TTS** for Indian languages, preserving native phonetic and prosodic characteristics.
> > >
> > > Overall, VAB’s construction relies on validated tool trajectories, culturally grounded query generation, multilingual quality filtering, and diverse speech synthesis. Detailed examples from benchmark and tool sets are provided in Appendices H and I to enable transparency. In addition, we will **open-source the benchmark, evaluation framework, and model responses** to enable full transparency and reproducibility.
> > >
> > > [1] BhashaKritika: Building Synthetic Pretraining Data at Scale for Indic Languages
> > >
> > > >**Q1:** Has any portion of the GPT-generated ground truth been validated by humans?
> > >
> > > We thank the reviewer for raising this point. To assess the quality of GPT-generated ground truth, we sampled 507 entries (~10% of the dataset) and divided them between two human annotators for verification. Of these, 496 entries were confirmed correct, yielding a **97.83%** validation accuracy.
> > >
> > > The few instances of partial errors were minor and did not compromise the overall quality of the benchmark. For example, in some location-based queries, the ground truth provided a broader region (e.g., *“Chennai”*) instead of a more specific locality (e.g., *“Velachery”*). We have added the details in Appendix L and also updated Section 3.2.2 to mention this validation.

---

> ### Author Response · Authors · 2025-11-21
>
> >**Q5:** Can you provide per-language breakdowns of failure types?
>
> We provide a detailed per-language evaluation of Tool Selection, Tool-Call Structure, and Parameter Filling in all task categories (Single, SinTC, Parallel, and Sequential) for Bengali, Malayalam, Marathi, Tamil, and Telugu in **Appendix Table 9**. This table reports evaluations per language and per model, enabling fine-grained comparison across Indic subsets. **Appendix E.1** presents a detailed analysis of the evaluation results across the languages. Some additional Indic language specific insights:
>
> - Malayalam is the most challenging language overall, showing the worst PF values amongst all the models.
> - Telugu and Bengali are the other languages that show poor performance, Telugu especially shows decent Parameter Filling scores for Single Tool Calling but falls sharply for Parallel Tool Calls.
> - Tamil shows the biggest drop in performance between SpeechLMs and ASR-LLMs.
>
> In addition, we also show some of the failure case examples in https://anonymous.4open.science/r/vab-2833/Failure_cases.md

---

> > ### Author Response · Authors · 2025-11-26
> >
> > Dear Reviewer,
> >
> > Thank you once again for your thoughtful and detailed review. Your feedback significantly strengthened our work particularly around inclusion of:
> > - human validation and agreement studies on LLM-generated ground truth
> > - assessment of GPT-4o-mini’s reliability as a judge
> >  - extension of multi-turn evaluation to Hindi
> > - significance tests for our main results.
> >
> > We wanted to ensure that all your concerns have been addressed.
> >
> > Thanks!

---

> ### Comment · Reviewer_Tfdb · 2025-11-26
>
> Hi authors, thank you so much for your detailed response. I am particularly encouraged to see the additional human agreement study, as I felt that that was the biggest weakness. This adequately addresses my concerns, as they established substantial agreement with humans with 2 metrics. I see value in a multilingual speech dataset of this scale, especially well positioned with agentic AI and safety, so I retain my positive assessment of this paper.

---

> > ### Author Response · Authors · 2025-11-28
> >
> > We sincerely thank the reviewer for the thoughtful follow-up and are very encouraged by their positive reassessment of the paper’s contributions, as well as the acknowledgement that we have fully resolved the earlier concerns.
> >
> > If the reviewer feels that the strengthened evidence and clarified methodology elevate the contribution beyond the initial evaluation, we would greatly appreciate their consideration in reflecting that through an increased score. We would be grateful to have their support in the overall decision process.

---

### Author Response · Authors · 2025-12-03
**Rebuttal Summary**

We thank all reviewers for their thoughtful comments and constructive feedback. We sincerely appreciate the reviewers for recognizing several strengths of our work, including:

- **Novelty and timeliness:** VoiceAgentBench is the first benchmark designed to evaluate the agentic capabilities of SpeechLLMs.
- **Broad task coverage:** including single-tool, multi-tool, sequential, multi-turn, safety-sensitive, multilingual, and culturally grounded evaluations.
- **Layered evaluation framework:** Tool Selection (TS), Tool Call Structure (TCS), Parameter Filling (PF), and Refusal Rate (RR), enabling fine-grained and interpretable diagnostics.
- **Extensive experimentation:** with thorough ablations and analyses supporting the central claims.

Their insights helped us improve the manuscript through additional analyses, experiments, and clarifications. Below, we summarize our responses and updates.

**Main Concerns Raised**

The primary concerns from reviewers were:
- Reliability of LLM-based judging and ground truth annotations
- Distribution and representation of Indian language/cultural contexts
- Use of synthetic speech (TTS)
- Significance testing and statistical confidence

**Key Experiments and Clarifications Added During Rebuttal**

During the rebuttal period, we conducted multiple additional analyses to address these concerns comprehensively. All results were positive and strengthened the paper. Notable updates include:

- **Human–LLM Agreement Study:** Human annotations showed substantial agreement with GPT-4o-mini as a judge. (Updated in Section 3.3 and Appendix M.)
- **Human Validation of Ground-Truth Annotations:** Achieved **97.8%** accuracy on a representative subset. (Updated in Section 3.2.2 and Appendix L.)
- **Clarification of Context & Language Distribution:** Added distribution analysis and a new figure illustrating language diversity in VoiceAgentBench. (Figure 4 in Appendix A)
- **TTS Quality Evaluation:** Included a pilot Mean Opinion Score (MOS) study showing TTS feasibility. (Appendix N)
- **Statistical Significance & Confidence Intervals:** Added McNemar’s tests and confidence intervals for main results, which reinforced the key claims. (Appendix E.2)
- **Random Sampling for Zero-Shot TTS Diversity:**  Added as a baseline to compare against our diversity-generation methods. (Appendix C and Sec. 3.2.4)
- **Extended Multi-Turn Hindi Evaluation:** Results aligned with the general trends reported in the paper. (Table 2 in Section 4)
- **Extended ASR-LLM Ablation:** Included IndicASR in the pipeline evaluation in the ablation. (Table 6 and Section 4.3)
- **Demo of model behaviors and failure cases:** Added an anonymized public demo showcasing examples and common failure patterns. Link: https://anonymous.4open.science/r/vab-2833/Examples.md

*Additionally, **Reviewer Tfdb** confirmed that their concerns had been thoroughly addressed in our rebuttal and upheld a positive overall assessment.*

------------------------------------------------------

**Broader Impact:**

VoiceAgentBench advances the development of speech-enabled agents by introducing the *first benchmark* centered on grounded reasoning and tool use directly from spoken input. By evaluating models not only for English but also across multilingual, multi-step, as well as safety-critical scenarios, it exposes systematic failure modes. Our benchmark encourages transparency, robustness, and responsible evaluation practices. Looking forward, we hope VoiceAgentBench:

- serves as a foundation for *reproducible and inclusive evaluation*, ultimately guiding progress toward more capable and reliable voice assistants.
- inspires *methodological innovations* in training multi-lingual SpeechLMs for complex, agentic, and safety-sensitive tasks.

---

### Note · Authors · 2026-01-27

I have read and agree with the venue's withdrawal policy on behalf of myself and my co-authors.

---

### Meta-Review · Area_Chair_iWj4 · 2026-01-03

**Summary:**

The reviewers’ concerns that informed my suggested decision for the paper are as follows: the work focuses heavily on the Indian accent, and the benchmark may not be sufficiently exhaustive due to the lack of validation in real-world settings.

**Reviewer Concerns:**

The authors have been able to address several reviewers' concerns in their rebuttal: (i) the potential introduction of hallucinations and inaccuracies due to the use of GPT-4o-mini instead of human annotators; (ii) the addition of a human agreement study on a representative subset of the benchmark; (iii) the lack of multi-turn evaluation in Indic languages, which was addressed through new experimental results; and  (iv) the inclusion of significance tests.

However, there are concerns that were not resolved or were insufficiently addressed include: (i) the relationship between this work and existing research on voice agents. Indeed, the authors themselves acknowledge that several related works were available prior to submission; (ii) the benchmark’s limitation to cascaded speech-to-text LLM (SpeechLM) systems; (iii) the omission of critical dimensions of agent performance from the evaluation, (iv)  the heavy focus on the Indian accent.  Furthermore, the benchmark may not be sufficiently exhaustive.

**Reviewer Scores:**

Tfdb confirms their score (6) in the rebuttal letter but might have increased it by +1.0.

nz5n would have confirmed their score as stated in the rebuttal letter

xvuT might have kept their score.

fPsi might have kept their score.

---

### Decision · Program_Chairs · 2026-01-26

Reject